# Pre-Sleep Low Glycemic Index Modified Starch Does Not Improve Next-Morning Fuel Selection or Running Performance in Male and Female Endurance Athletes

**DOI:** 10.3390/nu12092888

**Published:** 2020-09-22

**Authors:** Monique D. Dudar, Emilie D. Bode, Karly R. Fishkin, Rochelle A. Brown, Madeleine M. Carre, Noa R. Mills, Michael J. Ormsbee, Stephen J. Ives

**Affiliations:** 1Health and Human Physiological Sciences, Skidmore College, Saratoga Springs, NY 12866, USA; mdudar@skidmore.edu (M.D.D.); ebode@skidmore.edu (E.D.B.); kfishkin@skidmore.edu (K.R.F.); rbrown@skidmore.edu (R.A.B.); mcarre@skidmore.edu (M.M.C.); nmills@skidmore.edu (N.R.M.); 2Department of Nutrition, Food, and Exercise Sciences, Institute of Sport Sciences and Medicine, Florida State University, Tallahassee, FL 32306, USA; mormsbee@fsu.edu; 3Department of Biokinetics, Exercise and Leisure Sciences, School of Health Sciences, University of KwaZulu-Natal, Durban 4041, South Africa

**Keywords:** exercise, carbohydrates, time trial, substrate utilization, fat oxidation, fatigue, gastrointestinal distress, satiety

## Abstract

To determine the effects of pre-sleep supplementation with a novel low glycemic index (LGI) carbohydrate (CHO) on next-morning substrate utilization, gastrointestinal distress (GID), and endurance running performance (5-km time-trial, TT). Using a double-blind, randomized, placebo (PLA) controlled, crossover design, trained participants (*n* = 14; 28 ± 9 years, 8/6 male/female, 55 ± 7 mL/kg/min) consumed a LGI, high glycemic index (HGI), or 0 kcal PLA supplement ≥ 2 h after their last meal and <30 min prior to sleep. Upon arrival, resting energy expenditure (REE), substrate utilization, blood glucose, satiety, and GID were assessed. An incremental exercise test (IET) was performed at 55, 65, and 75% peak volume of oxygen consumption (VO_2peak_) with GID, rating of perceived exertion (RPE) and substrate utilization recorded each stage. Finally, participants completed the 5-km TT. There were no differences in any baseline measure. During IET, CHO utilization tended to be greater with LGI (PLA, 56 ± 11; HGI, 60 ± 14; LGI, 63 ± 14%, *p* = 0.16, *η*^2^ = 0.14). GID was unaffected by supplementation at any point (*p* > 0.05). Performance was also unaffected by supplement (PLA, 21.6 ± 9.5; HGI, 23.0 ± 7.8; LGI, 24.1 ± 4.5 min, *p* = 0.94, *η*^2^ = 0.01). Pre-sleep CHO supplementation did not affect next-morning resting metabolism, BG, GID, or 5-km TT performance. The trend towards higher CHO utilization during IET after pre-sleep LGI, suggests that such supplementation increases morning CHO availability.

## 1. Introduction

The importance of pre-exercise nutrition for exercise performance has been well documented [1,2,3,4,5,6]. However, given that many competitive endurance activities (training and/or competition) are scheduled early in the morning, there exists a major limitation: inadequate time in the morning prior to the event to properly fuel for sport. In addition, endurance athletes seldom consume much, if anything, before training or competitions of 75–90 min in duration [7,8,9,10,11]. Unfortunately, this behavior may result in sub-optimal physiological conditions such as carbohydrate depletion, dehydration, and fatigue [12], which will adversely impact training quality and performance. This issue highlights the need to develop strategies to provide adequate nutrition from foods, beverages, and/or supplements that athletes can consume pre-sleep without inducing gastrointestinal distress (GID) or disrupting normal sleep patterns [6,8]. As an intervention, the incorporation of a pre-sleep meal may provide an added “window of opportunity” for optimizing next-morning pre-race carbohydrate (CHO) availability and exercise performance.

Glucose is the body’s preferred energy substrate during endurance exercise. Currently, high glycemic index (HGI) CHOs are utilized by most athletes for pre- and/or intra-exercise nutrition due to their rapid breakdown which drastically increases blood glucose availability. However, complications such as GID, may arise with the consumption of HGI CHO sources prior to or during exercise due to the gastrointestinal sensitivity to nutrient intake [13]. Consequently, this raises the question as to the role of low glycemic index (LGI) CHO for exercise performance and nutrient timing. Contrary to a HGI, a LGI slowly digests carbohydrates, thus providing more stable and long-lasting glucose release, which may lower GID, both of which may better support endurance exercise performance [14].

Previous literature agrees that consuming LGI CHO prior to exercise results in enhanced fat oxidation [15,16,17,18], and likely improved exercise performance [14,19]. However, not all studies agree [20,21,22,23]. Stevenson and colleagues [22,23] investigated the effects of a low vs. high glycemic index evening meal, approximately 16 h prior (at 19:00 p.m.), on next-morning metabolic responses at rest and during exercise in males [23] and females [22]. Though in these studies [22,23] participants were fed a standard HGI breakfast (at 08:00 a.m.) three hours prior to a 60-min run at 65% VO_2max_. Although the breakfast elicited immediate post-prandial effects (lower glycemic and insulinemic responses), there was no significant effect of the previous evening’s dinner glycemic index on substrate utilization at rest or during running [22,23]. Though, given that meals were consumed 16 h prior to testing, and participants were given a standardized HGI breakfast in the morning, it is not surprising that any residual metabolic effects could not be detected. Given these limitations, it remains unknown if pre-sleep CHO supplementation can optimize next-morning endurance athlete fuel selection and performance, and whether low or high glycemic index would be preferential.

Though a LGI is touted as an efficacious source of CHO, a novel hydrothermally modified LGI starch supplement was developed to manage glucose levels by providing a slow and steady release of glucose to the body and brain for up to ten hours at a time [24]. In fact, data indicate a lower peak and less rapid rate of decline in blood glucose than conventional cornstarch, which is already considered a LGI CHO [25,26]. Currently, only a few studies have included the use of this novel LGI CHO supplement [15,17,21,27] all of which included supplement ingestion either before, during, or after exercise. No studies have investigated the effects of pre-sleep ingestion of a modified CHO on next-morning exercise metabolism and performance. However, there is a plethora of data investigating protein pre-sleep. The only study, to date, that investigated pre-sleep CHO-type beverage was conducted by Ormsbee et al. (2016) [28]. The authors investigated the effect of pre-sleep chocolate milk (HGI and protein) on endurance performance and found that chocolate milk resulted in increased carbohydrate oxidation in the morning, but effects did not translate to 10-km running performance improvements in females [28]. Given these data, we want to explore the effects of pre-sleep LGI CHO since it has the potential to have a positive impact based on the slow/long-lasting release of glucose, the lasting satiety, and if a HGI increases CHO oxidation, it is reasonable to suspect that LGI CHO would have the opposite effect, as noted in acute day of studies.

Accordingly, the purpose of this study was to determine the effects of nighttime pre-sleep supplementation with a novel LGI CHO on the next morning: (1) resting metabolism and GID; (2) metabolic and GID responses to incremental exercise; and (3) 5-km time trial running performance in trained male and female endurance athletes. It was hypothesized that the nighttime pre-sleep consumption of LGI CHO would, in the next morning, enhance fat utilization during exercise, decrease GID, and improve 5-km running time compared to a HGI CHO and a placebo (PLA) control.

## 2. Materials and Methods

### 2.1. Subjects

Trained male (*n* = 8) and female (*n* = 6) endurance runners between the ages of 18 and 45 years were recruited to participate in this study from local running clubs, triathlon teams, by word of mouth, flyers, and through an email distribution around the Skidmore College campus. Participants were included if they met the peak volume of oxygen consumption (VO_2peak_) qualifications (women: VO_2peak_ ≥ 40 mL·kg^−1^·min^−1^ and men: VO_2peak_ ≥ 45 mL·kg^−1^·min^−1^). Menstrual cycle status was recorded for all female participants, though they were scheduled independently of the menstrual cycle phase; notably, 3 out of 6 female participants did not have a menstrual cycle due to hormonal or contraceptive therapy. Participants were excluded if they smoked, had uncontrolled thyroid conditions, had been diagnosed with cardiac or metabolic disorders, regularly consumed anti-inflammatory drugs or any dietary supplements intended to improve performance, or had musculoskeletal injury that limited performance. All experimental procedures and risks of participation were explained verbally and in writing prior to participants providing written informed consent. Approval for this study was granted by the Human Subjects Institutional Review Board (IRB# 1901-786) of Skidmore College and is in accordance with the most recent revisions of the Declaration of Helsinki.

### 2.2. General Procedures

This was a double-blinded randomized placebo-controlled study included four total trials: one familiarization trial and three experimental trials. For the experimental trials, participants were randomly assigned to consume either (1) LGI, (2) HGI, or (3) PLA at least 2 h after their last meal and within 30 min prior to sleep on the evening before returning to the laboratory. Participants then arrived to the lab in the morning after an overnight fast (~7–9 h after supplement consumption) for an incremental exercise test (IET) and 5-km time trial (TT) (See Figure 1). Prior to the first experimental trial, participants were required to complete a one-day dietary food and exercise log. Participants were asked to replicate this diet and exercise, as closely as possible, prior to subsequent exercise trials. Participants abstained from the use of non-steroidal anti-inflammatory drugs, caffeine, alcohol, and/or vigorous activity at least 24 h prior to each experimental trial.

### 2.3. Supplementation

Over the span of the study, all participants were randomized to the order in which they received each of the following three supplements: (1) 532 mL of water mixed with 75 g of a HMS (LGI; Orange Flavor, SuperStarch^®^, The UCAN Co., Woodbridge, CT, USA) (270 kcal; 0 g PRO; 66 g CHO; 0 g FAT), (2) 532 mL of water mixed with 75 g of a HGI glucose-based supplement (Orange flavor, Gatorade^®^, PepsiCo, Inc., Purchase, NY, USA) (270 kcal; 0 g PRO; 67 g CHO; 0 g FAT), and (3) 532 mL of water mixed with a color and flavor-matched, non-nutritive PLA (PLA; Orange CRUSH flavor packet and Benefiber), with a volume of the powder visually similar to the other experimental conditions. Beverages were of similar taste, appearance, and consistency. The supplements were pre-packaged in inconspicuously labeled/coded opaque containers by a researcher not otherwise involved in the study.

### 2.4. Familiarization Protocol

Participants filled out a physical activity readiness questionnaire (PAR-Q), ACSM health preparticipation screening questionnaire, and a menstrual cycle history form (females only). Height was measured using a stadiometer (Seca 213, portable stadiometer, Chino, CA, USA), while body composition and weight were measured using air displacement plethysmography (BOD POD; COSMED, Chicago, IL, USA) [29].

Peak volume of oxygen consumption (VO_2peak_) testing was performed to assess baseline cardiorespiratory fitness and inclusion in the study. Gas exchange and ventilatory parameters were measured with a metabolic cart system (TrueOne 2400 Parvomedics, Salt Lake City, UT, USA) [30]. For each individual trial, the metabolic system was calibrated by a flow-calibration with a 3-L calibration syringe and gas analyzer calibration with gas mixture of known concentrations of oxygen (O_2_) and carbon dioxide (CO_2_) (16% O_2_; 4% CO_2_) according to manufacturer specifications, in addition to environmental data for standardization purposes. Participants were fitted with a nose clip, two-way non-rebreathe valve, and mouthpiece which was supported by a headpiece in order to collect expiratory gases for analysis by the metabolic cart. The VO_2peak_ protocol was performed on a treadmill (Woodway PPS Med, Waukesha, WI) and the protocol required a self-selected constant pace that was “comfortable but challenging.” Once the appropriate speed was determined, grade was increased at a rate of 2% every two minutes until the participant reached volitional fatigue [28]. During the last 15 s of each stage, HR was measured using a chest worn HR monitor (H7, Polar USA, Lake Success, NY, USA) and RPE was measured on a 1–10 categorical ratio scale.

### 2.5. Experimental Protocol

A recovery period of >72 h was required after the familiarization trial and between each testing day for all participants. On average, the time between trials was 192 ± 168 h. As sleep may have influenced exercise performance we asked participants to self-report their sleep duration prior to each visit. Participants then returned to the laboratory the following morning in a fasted, but well-hydrated, state between 05:30 and 08:30 a.m. Upon arrival, participants were asked to provide a urine sample to measure urine specific gravity using a hand-held refractometer to confirm hydration status [8]. Participants were provided with 250 mL of water to consume at their leisure before exercise and an additional 250 mL of water if their urine specific gravity indicated dehydration (>1.020). Thereafter, baseline measurements for height, weight, body composition, resting HR, satiety, GID, resting energy expenditure (REE), and capillary blood glucose (BG) were collected. GID and satiety were measured via a 100-mm visual analog scale (VAS) during baseline and one-minute post 5-km TT for each of the three experimental trials [31,32,33]. Each VAS scale was marked with “0 mm” (no GID; extreme hunger) and “100 mm” (extreme GID; extremely full) and participants were asked to draw a vertical line indicating their perceived GID and satiety accordingly. Both VAS and categorical scale have been documented in the literature as reliable perceptual measures of pain or discomfort [31,32,33].

REE was collected while participants rested quietly in a seated position with the headpiece and mask on for 15 min in a climate-controlled room with the metabolic cart system described above. Respiratory exchange ratio (RER) was recorded and relative substrate utilization (%FAT and %CHO) was estimated [34]. The last ten minutes were used for data analysis. Resting HR was then measured followed by blood sampling via finger stick. Capillary BG concentrations were measured using a commercially available glucometer (OneTouch Ultra 2 LifeScan, Milpitas, CA, USA) [35].

### 2.6. Incremental Exercise Test and 5-km Time Trial

Fifteen minutes following completion of baseline measurements, participants completed a three-minute warm up at a self-selected pace on the treadmill. Following the warm up, participants completed an incremental exercise test (IET) comprising of three stages of three minutes each at exercise intensities of 55, 65, and 75% of VO_2peak_ [34]. HR, RPE, and GID were recorded during the last 15 s of each three minute each stage. GID and RPE were measured during exercise using a categorical scale [31,32,33]. Upon completion of the IET, participants were given a five-minute active rest period where they were instructed to walk on the treadmill at a comfortable pace to allow their HR to return closer to baseline. Participants were allowed to use the restroom quickly as long as a researcher monitored them for safety. Following the active rest period, participants completed a 5-km TT. The TT was conducted rather than time-to-exhaustion to better mimic competition and pacing demands [36] and due to greater reliability in the repeatability of the results [37]. Participants were instructed to treat each TT as a competitive event and accordingly provide maximal effort. Participants could only see their distance during the TT and the time and speed were blinded. Additionally, participants ran both the IET and 5-km TT at 1% grade to best simulate the oxygen cost of outdoor running [38]. HR, RPE, and GID measurements were taken every 1 km. BG and HR were measured immediately post exercise and 10 min post exercise. HR, GID (VAS), and satiety (VAS) were also recorded one minute post 5-km TT.

### 2.7. Statistical Analysis

A sample size estimation was conducted (G*power, sample size estimator v.3.1.9.4; Kiel, Germany) for F-test family in a repeated measures design, using the following parameters: average effect size of night time supplementation of 0.39 on CHO oxidation [28], alpha level of 0.05, and minimal power of 0.8, which revealed a minimum sample size of 13 participants. Enrollment was targeted beyond this minimum in the possible event of dropout. All statistical analyses were performed using commercially available software (SPSS v.25, IBM, Armonk, NY, USA). A one-way repeated measures ANOVA was used to determine if differences existed at baseline across conditions (PLA, HGI, LGI). A two-way repeated measures ANOVA was used to analyze the potential impact of condition (PLA, HGI, LGI), exercise intensity (55, 65, 75% VO_2peak_), and their potential interaction on HR, RPE, GID, RER, % FAT and % CHO utilization. Two-way repeated measures ANOVA models were used to compare condition (PLA, HGI, LGI), distance (each km), and their potential interaction on RPE, HR, and GID during the 5-km TT. Lastly, a two-way ANOVA was used to determine potential differences across condition (PLA, HGI, LGI), time (baseline, one and ten minutes post 5-km TT), and their potential interaction on BG. Tests of normality were performed and Greenhouse–Geisser corrections were utilized if sphericity was violated. As men and women were recruited for this study, exploratory multi-variate ANOVA measures were conducted including sex as an independent covariate in the model for the above analyses. Significant main effects were followed up using Tukey’s Honestly Significant Difference, and *p* values were complemented by effect size, which in this model we used partial eta squared (*η*^2^). Alpha was set at 0.05. Data are presented as means ± standard deviation.

## 3. Results

### 3.1. Participants

Fourteen healthy endurance trained males (*n* = 8) and females (*n* = 6) completed all visits for this study. An overview of subject characteristics is presented in Table 1. There were no differences in self-reported sleep duration between visits (*p* = 0.56, *η*^2^ = 0.05, Table 1).

### 3.2. Effects of Supplement on Baseline Measures

Resting metabolic data are displayed in Table 2. There was no significant effect of supplement on baseline REE (PLA, 1689 ± 278; HGI, 1701 ± 308; LGI, 1732 ± 287 kcal·day^−1^, *p* = 0.72, *η*^2^ = 0.03; Table 2). There was a significant interaction of supplement and sex for baseline RER, and thus relative substrate utilization where males displayed a higher %FAT (PLA, 47.6 ± 5.4; HGI, 51.3 ± 11.5; LGI, 48.9 ± 11.8%, *p* = 0.02, *η*^2^ = 0.28) utilization compared to females (PLA, 46.9 ± 13.9; HGI, 28.3 ± 13.7; LGI, 34.5 ± 22.2%, *p* = 0.02, *η*^2^ = 0.28) at rest for HGI and LGI. However, all other baseline measures were unaffected by the supplement at baseline (all *p* > 0.05, Table 2).

### 3.3. Effects of Supplement on the Response to the Incremental Exercise Test (IET)

On average, during the IET, the LGI supplement tended to utilize less FAT (PLA, 44.1 ± 10.5; HGI, 39.7 ± 13.0; LGI, 37.5 ± 13.7%, *p* = 0.17, *η*^2^ = 0.14) and more CHO (PLA, 56.4 ± 10.6; HGI, 60.1 ± 14.3; LGI, 63.1 ± 13.9%, *p* = 0.17, *η*^2^ = 0.14; Figure 2) than the other two supplements, though this did not reach statistical significance. During the IET, there was no significant effect of supplement on VO_2_ (*p* = 0.23, *η*^2^ = 0.11, Figure 2C) or RER (*p* = 0.17, *η*^2^ = 0.14, Figure 2D). There was a tendency for an interaction of supplement with intensity for VO_2_ where values tended to be lower with the PLA during the lower intensity but equalized in the latter stages (*p* = 0.08, *η*^2^ = 0.18, Figure 2D). Expectedly, all metabolic parameters were significantly affected by exercise intensity (all *p* < 0.001, all *η*^2^ > 0.90, Figure 2A–D). The IET elicited an increase in GID (*p* = 0.04, *η*^2^ = 0.23, Figure 3) and RPE (*p* = 0.00, *η*^2^ = 1.00, data not shown). Supplementation had no effect on GID (*p* = 0.28, *η*^2^ = 0.10, Figure 3) or RPE (*p* = 0.55, *η*^2^ = 0.05, data not shown).

### 3.4. Effect of Supplement on 5-km TT

Supplement had no impact on HR during the 5-km TT (*p* = 0.89, *η*^2^ = 0.01, Figure 4A). Although there was a significant effect of running distance during the 5-km TT on GID (categorical scale) (*p* = 0.00, *η*^2^ = 0.58), there was no significant effect of supplement or an interaction of supplement by distance on GID (categorical scale) during the 5-km TT (Figure 4B). RPE was not impacted by supplement (*p* = 0.35, *η*^2^ = 0.01, Figure 4C). Running performance during the 5-km TT was unaffected by supplement (PLA, 21.6 ± 9.5; HGI, 23.0 ± 7.8; LGI, 24.1 ± 4.5 min, *p* = 0.94, *η*^2^ = 0.01, Figure 4D).

### 3.5. Effect of Supplement on Perceptual Responses of GID and Satiety to Exercise

Supplement had no significant effect on satiety from pre- to post-experimental trial (*p* = 0.39, *η*^2^ = 0.08; Figure 5A). There was no significant effect of supplement or time on pre- to post-experimental trial GID (VAS) (Figure 5B, *p* = 0.56, *η*^2^ = 0.03).

### 3.6. Blood Glucose (BG)

There were significant main effects for time (*p* = 0.00, *η*^2^ = 0.66), where blood glucose at baseline (PLA, 97.7 ± 8.1; HGI, 99.4 ± 8.8; LGI, 98.4 ± 9.3 mg·dL^−1^) was significantly increased immediately post-exercise (PLA, 127.2 ± 19.4; HGI, 131.0 ± 28.8; LGI, 124.4 ± 27.9 mg·dL^−1^, *p* = 0.00) and ten minutes post-exercise (PLA, 127.6 ± 23.9; HGI, 133.3 ± 24.6; LGI, 126.7 ± 23.3 mg·dL^−1^, *p* = 0.00; Figure 6), but no differences were observed between immediate and ten minutes post exercise. There were no significant differences in BG between supplements (*p* = 0.54, *η*^2^ = 0.04) at any time point or an interaction (*p* = 0.87, *η*^2^ = 0.02).

## 4. Discussion

The present study is the first to assess the effects of pre-sleep supplementation with a novel LGI CHO as compared to HGI CHO or placebo control on next-morning (~8 h later) exercise metabolism, GID, and endurance performance in male and female endurance athletes. It was hypothesized that the nighttime pre-sleep consumption of LGI CHOs would increase fat utilization during morning exercise, decrease GID, and improve 5-km TT performance. The primary findings were as follows: (1) supplementation had no significant effect on REE, CHO, or FAT utilization at rest, though females tended to utilize more CHO in the HGI and LGI supplement at rest; (2) supplementation had no significant effect on substrate utilization during graded submaximal exercise; (3) blood glucose was not different among supplements at any point during the trial; (4) perceptions of GID were not different among supplements; (5) supplementation had no discernable significant effect on 5-km TT performance. Although our data do not support our original hypothesis, the present study suggests that there are no detrimental effects of supplementing with either LGI or HGI CHO pre-sleep in endurance athletes and thus, they may be utilized as a feeding window and fueling strategy to ingest adequate daily energy intake.

The gastrointestinal tract can be very sensitive to the foods and beverages we consume. Unfortunately, nutrient ingestion prior to and during exercise may lead to GID. Baur and colleagues (2016) reported that GID increased after the consumption of the same hydrothermally modified starch (HMS) LGI supplement that our current study used [15]. Baur et al. (2016) compared the HMS to an HGI CHO supplement when ingested prior to, and during, prolonged cycling in ten trained male cyclists and triathletes [15]. It was reported that there were likely large correlations between mean sprint nausea (*r* = −0.51) and total GID (*r* = −0.53) and exercise trial, showing that GID contributed to reduced cycling performance [15]. Further, there was a HMS-associated increase in GID negatively effecting sprint cycling performance [15]. Given that HMS is slow releasing under normal digestion supplements, malabsorption may be the explanation for the primary pathophysiologic mechanism of LGI CHO-induced GID during exercise. Unlike the findings of Baur et al. (2016), the present study found no effect (positive or negative) on GID and performance. Perhaps the pre-sleep ingestion of LGI CHO avoids the LGI CHO-induced increase in GID in morning endurance performance. This is likely because the body can digest the LGI CHO during the overnight period. Participants in the Baur et al. (2016) study consumed LGI CHO during the exercise as well, which likely caused the incidences of GID with HMS ingestion [15].

An LGI CHO may still be an optimal source of CHO for athletes given its previously reported low osmolality, low insulin impact, slow release factor, and maintenance of blood glucose levels [5,24,25,26]. In general, elevated insulin levels attenuate lipolysis and fat metabolism, thus increasing utilization of CHO. Even though it is well documented that consuming LGI carbohydrates before exercising results in enhanced fat oxidation, or at least maintaining euglycemia during exercise [17,18,21,39,40,41,42,43], and possibly improved performance [44], though not all agree [16,20]. Data from the present study, albeit in a different methodological approach, do not support these findings, as we found no effect of LGI CHO, or HGI CHO for that matter. When comparing LGI to HGI, some studies have reported enhanced exercise performance [19,39,43,45,46] while other studies report no differences [41,42,47,48,49,50]. For example, Baur et al. (2016) reported an increase in total FAT oxidation and reduction in CHO oxidation with LGI supplementation 30 min before as well as during exercise [15], which disagrees with the findings of the present study utilizing pre-sleep supplementation of LGI. These inconsistencies may be explained by, principally, time but other methodological differences, such as timing or dose of CHO supplementation, type of exercise protocol (i.e., cycling versus running), or sample size should also be considered. Researchers have reported muscle glycogen sparing with LGI compared to HGI CHO [47], which may be explained by improved fat oxidation. Our findings are in accordance with previous literature that LGI and HGI CHO do not improve running TT performances [21,28].

Glycemic control is extremely important for those training and competing in endurance competitions and increasing fat oxidation could potentially benefit performance by preserving glycogen stores [51]. To maximize glucose fueling, the timing of pre-exercise consumption of CHO is essential, along with the type/amount of exercise being performed. The time of consumption may alter the metabolic effects. Studies have shown that CHO consumed one to four hours prior to exercise resulted in a decline in glucose and insulin basal levels prior to exercise [2,52]. Further research has reported that CHO consumed ≤ 60 min before exercise leads to elevated blood glucose and insulin levels immediately prior to exercise [47,53,54,55]. These findings emphasize the importance of nutrient timing and the exploring how the body performs from nutrient consumption solely the night before exercise takes place.

The trend towards higher CHO utilization during exercise after pre-sleep consumption of HGI or LGI CHO, perhaps more so in LGI, might suggest that pre-sleep LGI CHO supplementation increases morning CHO availability or more stable bioavailability, though more research is needed as this was not directly investigated in the present study. Due to the exercise paradigm used in the current study, the 5-km TT run lasting ~20–30 min could present itself as a higher intensity glycolytic exercise than longer endurance exercise performance trials. Research on the effects of CHO feeding for endurance exercise indicates that some measures of performance are more sensitive than others, and short duration exercises may not be long enough to cause CHO depletion and reveal potential effects of pre-sleep CHO supplementation [12]. This might explain the insignificant differences in 5-km TT performance in the current study, and perhaps longer bouts, and/or larger sample sizes, are required to reveal an effect. There was, however, a significant effect between supplement and sex for resting CHO and FAT oxidation in this study, where females utilized more CHO with LGI and HGI (PLA was consistent between sexes). This suggests females resting fuel selection may respond differently to pre-sleep LGI or HGI CHO supplementation, but further work is needed.

In the present study, which utilized a graded and shorter duration endurance event, we found no benefit with pre-sleep ingestion on enhancing exercise performance. A contributing factor for the lack of significant positive impact on exercise performance may be attributed to the relatively short duration of the exercise stimulus incorporated in the present study [12], the amount of CHO, and/or sample size. When exercise is prolonged in a moderately intense state, CHO oxidation gradually decreases while fat oxidation increases [51,56]. Muscle glycogen utilization decreases due to reduced muscle glycogen availability [57] hence why CHO supplementation is vital for exercise of longer duration since the body relies on CHO as fuel [13]. The exercise module that was used in the present study was based on previous literature that found an effect of nighttime feeding altering morning metabolism in a 10-km run [57], and was preceded by an incremental exercise trial of three five-minute stages at 55, 65, and 75% VO_2peak_ [57]. That protocol was altered to test a 5-km timed trial with an IET comprised of three three-minute stages at the same intensities. A main reason why those times were chosen include efficiency and time restraints. Additionally, not measuring substrate utilization during the 5-km TT limited the current study’s understanding of substrate metabolism to only the initial nine-minute incremental test but this was intentional to allow the athletes to give their best efforts and be minimally distracted. Contrary to our hypothesis, we found that pre-sleep supplementation with LGI CHO tended to the lowest FAT oxidation as compared to HGI and placebo control. In the present study, we cannot ascertain the mechanisms responsible such as altered intramuscular CHO availability, or altered bloods level of glucoregulatory hormones (i.e., insulin and glucagon).

### Experimental Considerations

Future studies should consider measuring exercise performance in live race scenarios, such as overland 5-km running events with performance feedback, for longer duration endurance bouts (e.g., 10 km, half-, or full-marathon), and explore optimal dosing strategies. Additionally, future work should determine if CHO availability is altered with pre-sleep CHO feeding by examining muscle glycogen, and with further consideration for sex differences, as females were shown to have higher CHO utilization than males at rest following both HGI and LGI pre-sleep supplementation. This observation contrasts with relatively established findings, but several factors could have contributed to females utilizing more CHO in the morning; we would like to acknowledge that the study was not designed to test sex differences and there were fewer female participants (*n* = 6) and larger studies may prove otherwise. Females also had, on average, lower VO_2peak_ value (49.9 ± 4.3 mL/kg/min vs. males at 59.5 ± 5.5 mL/kg/min), and lower body weight and thus higher relative CHO loading and thus fitness level and body weight may play a role. Another consideration of this study could be the dose of CHO that was administered; 66 g of CHO may not be enough to last the ~eight hours to the exercise trial. Future studies should investigate different dosages, dosing approaches (e.g., g/kg), and/or timings of nighttime CHO supplementation for next-morning endurance performance in a larger sample, with measurements of circulating glucoregulatory hormones or muscle glycogen which could provide greater mechanistic insight.

## 5. Conclusions

The present study is the first to assess the effects of pre-sleep LGI versus HGI CHO supplementation on next-morning exercise metabolism, GID, and endurance performance in male and female endurance athletes. The data indicate that pre-sleep supplementation with LGI, HGI, or PLA did not differ in GID response during exercise. There were no differences between supplements for resting REE or RER, BG, or TT performance. In a secondary analysis, there was an interaction of supplement and sex for FAT and CHO utilization at baseline with females utilizing more CHO with the pre-sleep LGI and HGI, which should be explored further. In this study, consuming a CHO supplement pre-sleep, and not within a couple of hours of exercise, might reduce GID, allowing for adequate digestion and absorption. Future studies should investigate the effect of pre-sleep CHO supplementation on the endurance performance of the following morning.

## Figures and Tables

**Figure 1 nutrients-12-02888-f001:**
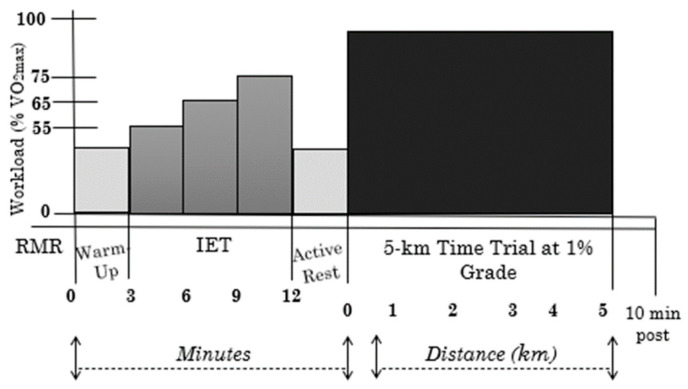
Experimental Overview; RMR = resting metabolic rate; IET: incremental exercise test.

**Figure 2 nutrients-12-02888-f002:**
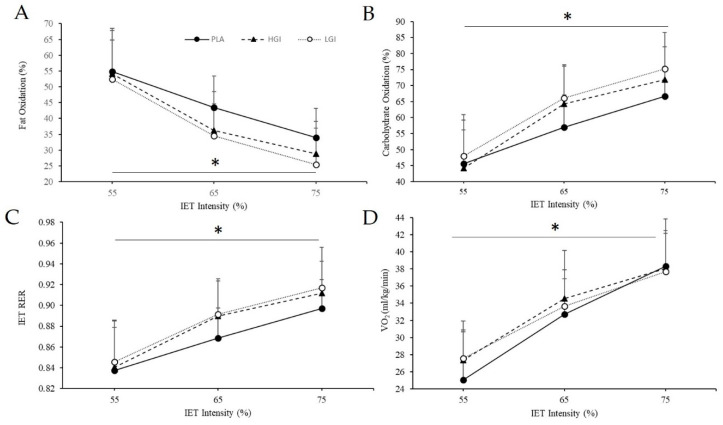
Metabolic Response to incremental exercise test (IET) at 55, 65, and 75% of VO_2peak_ between placebo (PLA), high glycemic index (HGI), and low glycemic index (LGI) supplements (*n* = 14). (**A**) Relative fat utilization (%FAT), (**B**) relative carbohydrate utilization (%CHO), (**C**) respiratory exchange ratio, and (**D**) VO_2_. Data expressed as means ± SD. * effect of intensity, *p* < 0.001.

**Figure 3 nutrients-12-02888-f003:**
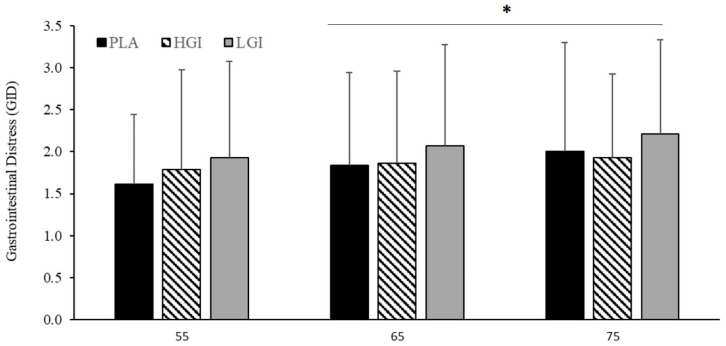
Gastrointestinal distress (GID; categorical scale) during incremental exercise trial (IET) at 55, 65, and 75% of VO_2peak_ (*n* = 14) between placebo (PLA), high glycemic index (HGI), and low glycemic index (LGI) supplements. Data expressed as means ± SD. * effect of intensity, *p* = 0.04.

**Figure 4 nutrients-12-02888-f004:**
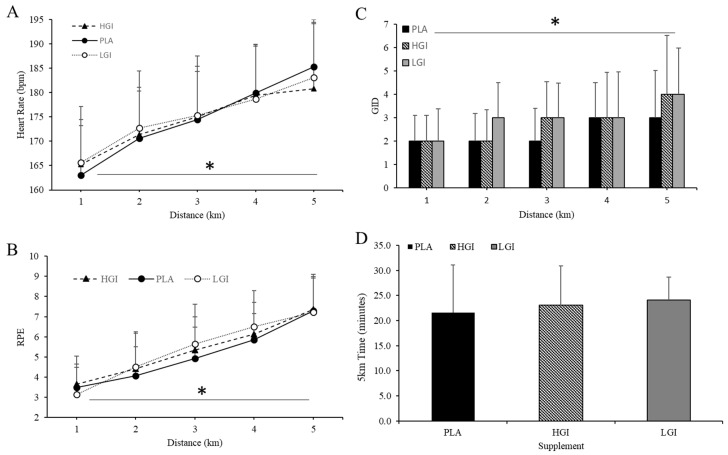
(**A**) Heart rate (HR), (**B**) gastrointestinal distress (GID, CS), (**C**) rating of perceived exertion (RPE) and (**D**) time (min) for 5-km time trial performance between placebo (PLA), high glycemic index (HGI), and low glycemic index (LGI) supplements (*n* = 14). Data expressed as means ± SD. * significant effect for distance *p* < 0.05.

**Figure 5 nutrients-12-02888-f005:**
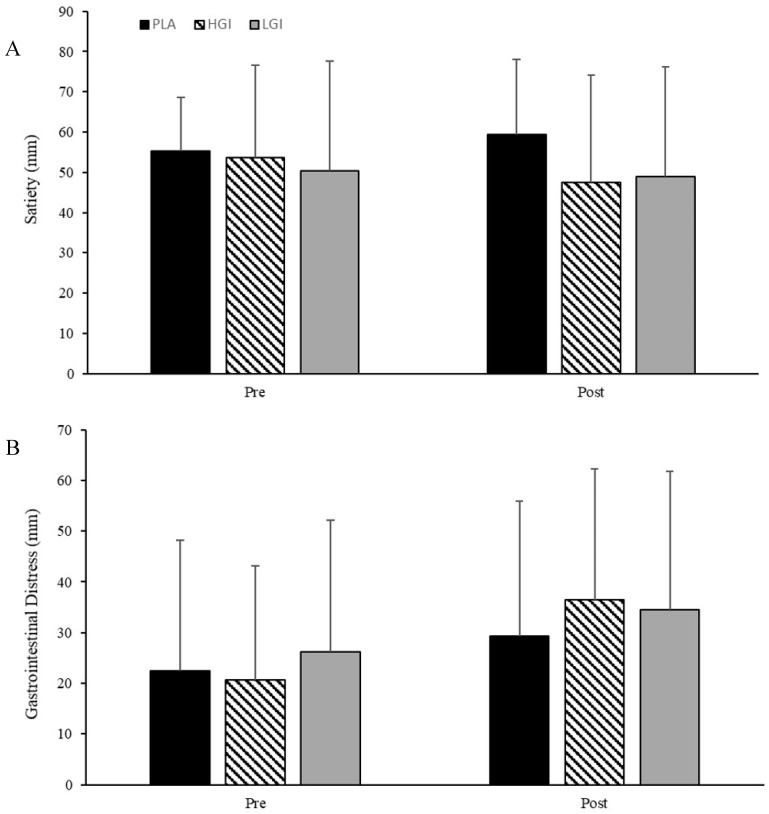
(**A**) Satiety (mm; VAS) and (**B**) gastrointestinal distress (GID; mm; VAS) at baseline (pre) and at conclusion (post) of running the 5-km time trial performance between placebo (PLA), high glycemic index (HGI), and low glycemic index (LGI) supplements (*n* = 14). Data expressed as means ± SD.

**Figure 6 nutrients-12-02888-f006:**
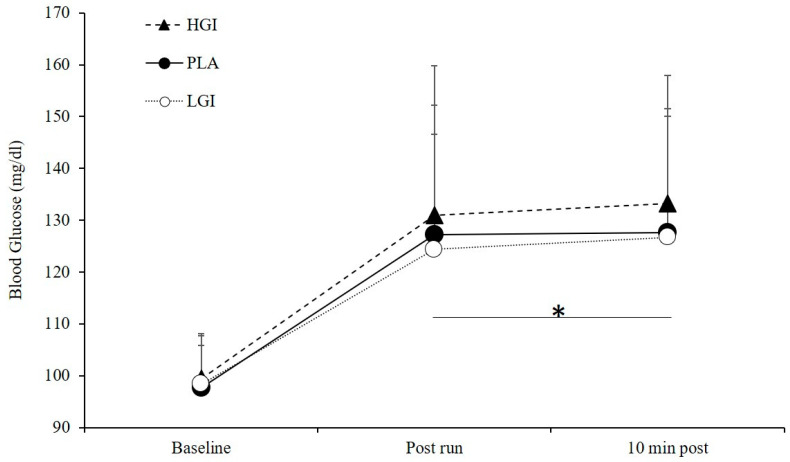
Blood glucose levels (mg·dL^−1^) at baseline, 1 min post 5-km TT run, and 10 min post 5-km TT run trial between placebo (PLA), high glycemic index (HGI), and low glycemic index (LGI) supplements (*n* = 14). Data expressed as means ± SD. * significant effect of time, *p* = 0.00.

**Table 1 nutrients-12-02888-t001:** Subject Characteristics.

Characteristic	Combined	Male	Female
Sex (*n*, M/F)	14	8	6
Age (years)	28 ± 9	29 ± 9	27 ± 10
Height (cm)	169.3 ± 10.4	176.0 ± 7.1	160.4 ± 6.8 *
Weight (kg)	64.3 ± 9.8	70.1 ± 7.8	56.5 ± 6.2 *
Body fat (%)	18.9 ± 5.6	14.7 ± 3.5	23.9 ± 2.5 *
VO_2peak_ (mL·kg^−1^·min^−1^)	55.4 ± 6.9	59.5 ± 5.5	49.9 ± 4.3 *

**Note:** Data expressed as means ± SD. VO_2peak_, peak oxygen uptake. * *p* < 0.05 male vs. female.

**Table 2 nutrients-12-02888-t002:** Baseline Measurements.

Supplement				
Variable	PLA	HGI	LGI	*p* Value
Visual Analogue Scale (VAS) (mm)				
Gastrointestinal Distress (GID)	22.4 ± 25.8	20.6 ± 22.5	26.2 ± 26.0	0.59
Satiety	55.3 ± 13.3	53.6 ± 23.0	50.3 ± 27.2	0.73
Substrate Oxidation				
FAT (%)	47.3 ± 9.5	41.4 ± 16.8	42.7 ± 17.8	0.16
Carbohydrate (CHO) (%)	53.2 ± 9.6	59.1 ± 16.9	57.8 ± 18.0	0.16
VO_2_ (mL/kg/min)	3.9 ± 0.4	3.8 ± 0.4	3.8 ± 0.4	0.84
Resting Energy Expenditure (REE) (kcal·day^−1^)	1689 ± 278	1701 ± 308	1732 ± 287	0.72
HR (bpm)	57.7 ± 8.8	57.3 ± 10.6	59.9 ± 10.0	0.06
Blood Glucose (BG) (mg·dL^−1^)	97.7 ± 8.1	99.4 ± 8.8	98.4 ± 9.3	0.85
Urine Specific Gravity (USG) (a.u.)	1.02 ± 0.01	1.02 ± 0.01	1.02 ± 0.01	0.91
Sleep (h)	7.2 ± 0.8	7.2 ± 0.9	6.9 ± 1.2	0.68

**Note:** Data are means ± SD. PLA: placebo; HGI: high glycemic index; LGI: low glycemic index; VAS: visual analogue scale; GID: gastrointestinal distress; REE: resting energy expenditure; BG: blood glucose; USG: urine specific gravity. Data expressed as means ± SD.

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
