# Peer review of "Pre-Sleep Low Glycemic Index Modified Starch Does Not Improve Next-Morning Fuel Selection or Running Performance in Male and Female Endurance Athletes"

_nutrients, 2020, doi:10.3390/nu12092888_

Round 1

Reviewer 1 Report

The authors present a study in which they intend to determine the effects of pre-sleep supplementation with a novel low glycemic index carbohydrate on next morning substrate utilization, gastrointestinal distress and endurance running performance (5km time-trial).
The work is interesting and innovative, however the sample size is somewhat reduced, if it were larger the results would have another consistency. Nevertheless, it is well structured and reasoned, with some items to be corrected.

Many acronyms appear in the summary and throughout the work, sometimes making the reading unclear. Some acronyms are not defined the first time they appear, others have no definition and others are only defined later in the text.

Abstract

The abstract has more than 200 words, which is not in accordance with the rules.

Key words

Some keywords not found in MeSH.

Material and methods

Should it be specified whether the return to the participants' laboratory was the same for everyone? If even transport and distance? Or were they all in the same place in equal conditions? Because this conditions imply energy expenses.

Throughout the work, reference is made to sleep, sleep quality being an important variable for our metabolism, and having not been monitored, could this not have any influence on the results?

Author Response

The authors present a study in which they intend to determine the effects of pre-sleep supplementation with a novel low glycemic index carbohydrate on next morning substrate utilization, gastrointestinal distress and endurance running performance (5km time-trial).
The work is interesting and innovative, however the sample size is somewhat reduced, if it were larger the results would have another consistency. Nevertheless, it is well structured and reasoned, with some items to be corrected.

RESPONSE: We sincerely thank the reviewer for their time, support for the project, and constructive comments, which we have responded to in a point-by-point fashion below.

Many acronyms appear in the summary and throughout the work, sometimes making the reading unclear. Some acronyms are not defined the first time they appear, others have no definition and others are only defined later in the text.

RESPONSE: We have reviewed and revised the manuscript for acronym use and have made corresponding changes.

 Abstract

The abstract has more than 200 words, which is not in accordance with the rules.

RESPONSE: Thank you, we apologize as we typically format for Pubmed (250 words), we have since changed the abstract to fit within the Journal guidelines (current word count = 199).

 Key words

Some keywords not found in MeSH.

RESPONSE: According to the Journals directions to authors specifies “Three to ten pertinent keywords need to be added after the abstract. We recommend that the keywords are specific to the article, yet reasonably common within the subject discipline.”. Thus, we feel as though the keywords fit within the recommendation outlined above.

Material and methods

Should it be specified whether the return to the participants' laboratory was the same for everyone? If even transport and distance? Or were they all in the same place in equal conditions? Because this conditions imply energy expenses.

RESPONSE: This is an interesting question. Each participant whilst having their own living arrangements (some on campus athletes, others off campus), under each condition participant living arrangements and transport to the facility were matched, thus energy expenditure to the facility is first matched between conditions and given the proximity of parking and small campus any such energy expenditure is minimal particularly given the fitness level of the individual.

Throughout the work, reference is made to sleep, sleep quality being an important variable for our metabolism, and having not been monitored, could this not have any influence on the results?

RESPONSE: We thank the reviewer for highlighting this, as we agree sleep was a potentially important moderator, we actually did assess self-reported sleep, and reported these data in Table 2. We realized this was missing in the methods and have since added this measure to this section and added text report to the results section. Briefly, to clarify there were no differences in self-reported sleep duration between visits (PLA, 7.2±0.8; HGI, 7.2±0.9; LGI, 6.9±1.2 hours, p = 0.56, Æž2 = 0.05).

Author Response

We sincerely thank the reviewer for their time, support for the project, and constructive comments, which we have replied in a point-by-point fashion in the PDF document. Changes made to the document are highlighted red.

Reviewer 3 Report

Obviously, this is a novel idea for research and the protocol is well designed and structured

Additionally, as you refer to your paper, though without statistical significance, prior to sleep carbohydrate ingestion might play a vital role for early morning exercise.

My minor revision comments would be explained as follows:

  • You need to consider the small number of your sample, take under consideration the range of age that could affect the metabolic profile of the glucose uptake (38 the older male and 17 the younger female)
  • It would be more helpful to standardize the carbohydrate ingestion expressed as gr per Kg of body weight. A standard quantity of carbohydrate ingestion might affect differently a low body weight comparing to a higher
  • Due to the intensity of the exercise protocol, stress hormones could manipulate the blood glucose levels and measuring relative hormone profile could support your hypothesis even better

Nevertheless, it is a very well-designed protocol and besides the above comments, that, in my opinion, should be taken into account in your experimental considerations, your paper provides data that will help sport nutrition research and exercise performance enhancement.

Author Response

Obviously, this is a novel idea for research and the protocol is well designed and structured

Additionally, as you refer to your paper, though without statistical significance, prior to sleep carbohydrate ingestion might play a vital role for early morning exercise.

RESPONSE: We sincerely thank the reviewer for their time, support for the project, and constructive comments, which we have responded to in a point-by-point fashion below.

My minor revision comments would be explained as follows:

  • You need to consider the small number of your sample, take under consideration the range of age that could affect the metabolic profile of the glucose uptake (38 the older male and 17 the younger female)

RESPONSE: We appreciate this concern, as biological aging is an important interest; however, only with more advanced levels of aging, such as over 60 years of age (e.g. see https://www.ncbi.nlm.nih.gov/pmc/articles/PMC3664017/ ), are there any pronounced issues with glucose handling. Moreover, the relatively modest age (~30) coupled with the high activity and fitness levels of the individuals (see https://care.diabetesjournals.org/content/39/11/2065 ), we do not believe that the ages we have in the study pose an issue. Lastly, these ages are highly typical and representative of endurance athletes. Thus, given these considerations, and absence of overt metabolic disease, combined with the within subjects crossover design, we do not believe age was a mitigating factor in our relatively young cohort.

  • It would be more helpful to standardize the carbohydrate ingestion expressed as gr per Kg of body weight. A standard quantity of carbohydrate ingestion might affect differently a low body weight comparing to a higher.

RESPONSE: The reviewer brings up an interesting point. We obviously did not design the study in this way, but aimed to follow the way people may typically consume supplements, that is in an absolute dosing approach. However, we calculated the relative CHO loading and the average was 1.02 ± 0.16 g/kg CHO, ½ to 1/10 of the values used for chronic CHO loading and on the relatively low end of the 1-4 g/kg immediately prior to exercise (see https://www.ncbi.nlm.nih.gov/pmc/articles/PMC6566225/ ). We simply do not have adequate range of body weights (SD in relative dosing was 0.16 g/kg) or a serious bimodal distribution of body weights, with enough statistical power to fully address this question. Though we agree that future studies should explore dosing and using a relative dosing approach may prove beneficial, and we have added discussion of this point to the experimental considerations section of the discussion in the paper.

  • Due to the intensity of the exercise protocol, stress hormones could manipulate the blood glucose levels and measuring relative hormone profile could support your hypothesis even better

RESPONSE: The reviewer brings up an excellent point, and we agree that measurement of circulating glucoregulatory hormones would enhance the mechanistic insight to the potential impact of pre-sleep CHO ingestion on next morning physiology and performance. However, that was not possible in the present study, due to institutional restraints. Though we believe this is an interesting point we have since added this to the experimental considerations section of the discussion. Thank you.

Nevertheless, it is a very well-designed protocol and besides the above comments, that, in my opinion, should be taken into account in your experimental considerations, your paper provides data that will help sport nutrition research and exercise performance enhancement.

RESPONSE: We sincerely appreciate the reviewers feedback and positive view of the paper, we have since added points raised above into the experimental considerations section. Thank you.

Round 2

Reviewer 2 Report

The authors have improved the manuscript significantly and responded well to my concerns. Can the authors add a column of ANOVA P for table 2 and format references?

Reference 2 and many other subsequent references require formatting. Some of them have the first letter in upper case some of them are all lower cases. Please be consistent.

Author Response

The authors have improved the manuscript significantly and responded well to my concerns. Can the authors add a column of ANOVA P for table 2 and format references?

RESPONSE: We appreciate the reviewers' recognition of our efforts to improve the paper. We have added the p-value column with values in table 2.

Reference 2 and many other subsequent references require formatting. Some of them have the first letter in upper case some of them are all lower cases. Please be consistent.

RESPONSE: We apologize for the inconsistencies in the references, we have edited these, thank you.